# Building a community-based open harmonised reference data repository for global crop mapping

Hendrik Boogaard[1]*, Arun Kumar Pratihast[1], Juan Carlos Laso Bayas[2], Santosh Karanam[2], Steffen Fritz[2], Kristof Van Tricht[3], Jeroen Degerickx[3], Sven Gilliams[3]

1 Wageningen Environmental Research (WENR), Wageningen University & Research, Wageningen, Netherlands, 2 International Institute for Applied Systems Analysis (IIASA), Laxenburg, Austria, 3 Vlaamse Instelling Technologisch Onderzoek (VITO), Mol, Belgium

* hendrik.boogaard@wur.nl

**Data Availability Statement:** The reference data are held in a public repository (Zenodo) under the following DOIs: ○ 10.5281/zenodo.7609546 ○ 10.5281/zenodo.7609500 ○ 10.5281/zenodo.7593734

## Abstract

Reference data is key to produce reliable crop type and cropland maps. Although research projects, national and international programs as well as local initiatives constantly gather crop related reference data, finding, collecting, and harmonizing data from different sources is a challenging task. Furthermore, ethical, legal, and consent-related restrictions associated with data sharing represent a common dilemma faced by international research projects. We address these dilemmas by building a community-based, open, harmonised reference data repository at global extent, ready for model training or product validation. Our repository contains data from different sources such as the Group on Earth Observations Global Agricultural Monitoring Initiative (GEOGLAM) Joint Experiment for Crop Assessment and Monitoring (JECAM) sites, the Radiant MLHub, the Future Harvest (CGIAR) centers, the National Aeronautics and Space Administration Food Security and Agriculture Program (NASA Harvest), the International Institute for Applied Systems Analysis (IIASA) citizen science platforms (LACO-Wiki and Geo-Wiki), as well as from individual project contributions. Data of 2016 onwards were collected, harmonised, and annotated. The data sets spatial, temporal, and thematic quality were assessed applying rules developed in this research. Currently, the repository holds around 75 million harmonised observations with standardized metadata of which a large share is available to the public. The repository, funded by ESA through the WorldCereal project, can be used for either the calibration of image classification deep learning algorithms or the validation of Earth Observation generated products, such as global cropland extent and maize and wheat maps. We recommend continuing and institutionalizing this reference data initiative e.g. through GEOGLAM, and encouraging the community to publish land cover and crop type data following the open science and open data principles.

**Funding:** The work has received funding from the European Space Agency (ESA) WorldCereal project under the project number No. 4000130569/20/I-NB and by the Open Earth Monitor (OEMF) project, a project funded by the European Union's Horizon Europe research and innovation programme under grant agreement No.-101059548. ESA defined the general WorldCereal project specifications which included the set-up and building of a reference data repository as one of the main components of the WorldCereal system.

**Competing interests:** The authors have declared that no competing interests exist.

## 1. Introduction

In recent years, there has been a significant growth in the use of open source satellite data repositories for global ecosystem mapping (e.g. geographic distribution of crop areas [1]; land cover map [2], forest cover maps [3], biodiversity maps [4] etc.). All satellite data analysis requires ground truth data (further referred to as reference data) to produce reliable products and services [5–7]. With the rise of crowdsourcing [8, 9], the proliferation of open data platforms [10] and FAIRification process [11, 12], these data are now publicly available for satellite interpretation [13]. There are an increasing number of researchers that make use of existing reference data for satellite-based models/algorithms calibration and validation. Examples are characterising forest change [14], global cropland and field size mapping [15], agricultural land use mapping [16].

Reliable crop type and cropland maps are required to create policy-relevant information in support of monitoring and enhancing sustainable food production [7, 17]. However, the availability of reference datasets are the current major hurdle to produce these maps [7]. Several initiatives, including the Group on Earth Observations Global Agricultural Monitoring Initiative (GEOGLAM) Joint Experiment for Crop Assessment and Monitoring (JECAM) sites, the International Institute for Applied Systems Analysis (IIASA) citizen science platforms (LACO-Wiki, Geo-Wiki), the Radiant MLHub, the Future Harvest (CGIAR) centres, the National Aeronautics and Space Administration Food Security and Agriculture Program (NASA Harvest); are stepping up to tackle this hurdle for agricultural monitoring.

The GEOGLAM-JECAM initiative is mainly focused on collaboration, networking, and data/method sharing for crop area, condition monitoring and yield estimation, but unfortunately to date, these initiatives have no open data sharing portal.

IIASA's citizen science platforms (Geo-Wiki, LACO-Wiki) host a large number of reference datasets [18, 19]. These datasets have been extensively used for land cover mapping. One advantage is that these datasets are gathered through crowdsourcing efforts that use high-resolution satellite imagery. The downside of these data sets is that their quality varies depending on purpose (experts/volunteers) and knowledge of those contributing. Furthermore, the data stored in these platforms comes from projects undertaken in the past, and many of the data sets contain relatively broad land cover classes, thus being less suitable for annual crop land and crop type mapping [19].

The Global Agricultural Research Data Innovation & Acceleration Network (GARDIAN [20]) is a Future Harvest (CGIAR) initiative to discover publications and datasets from data repositories across all CGIAR Centers. These repositories hold some useful open project data sets but in various formats and descriptions that need to be interpreted by consulting the associated peer-reviewed research papers.

NASA Harvest recently published the CropHarvest dataset, a crop dataset of geo-referenced labels with satellite data inputs, each consisting of latitude, longitude, the associated crop type label, and a satellite pixel time series [21]. It includes several data sets in one structure but lacks a standardised crop and land cover description and metadata is limited to a reference.

The Radiant MLHub is a new addition that delivers reference data for machine learning algorithm training and validation [22]. This platform provides original datasets, models and metadata connecting models. The datasets lack comprehensive global coverage, one common structure and sometimes geolocations are altered (for data privacy reasons), making them challenging to utilise in other projects.

As a result, the operational use of these data repositories to produce reliable crop type and cropland maps at global scale is limited due to the following factors: 1) lack of data accessibility and formats, 2) inadequate data standardisation, 3) unknown data quality, 4) inconsistent

and/or incomplete metadata, 5) limited spatial, temporal, and thematic coverage, and 6) unclear (re)use data policies. In summary, existing data repositories are unable to supply globally harmonised reference data for EO model training and product validation.

Hence, there is an urgent need for a global, extensive open reference data repository with recent data on crop types and land cover. The research was conducted within the framework of a European Space Agency (ESA) funded WorldCereal project. WorldCereal developed an EO based system for timely global crop monitoring at field scale including harmonised reference data for training or validation. Compared to existing data repositories, our WorldCereal reference data repository offers some immediate advantages:

1. Global comprehensive community-based open reference data repository for crop type and cropland mapping, harmonised and with complete metadata and supporting documents (e.g. data formats, legends, observation date assessment, metadata).

2. User-friendly stepwise harmonization protocol to further enlarge the data set and stimulate the community to share data and build trust for long-term sustainability.

The objective of the WorldCereal reference data repository is:

1. Discover data sharing institutions across the world and make an overview of existing reference data with a focus on cereal crops.

2. Collect the data, develop, and apply a data curation and harmonisation protocol for these heterogeneous reference data.

3. Assess the fitness for use of these data through spatial, temporal, and thematic accuracy rules.

4. Provide and contribute harmonised data sets access through a user-friendly web interface and build the trust and long-term relationships with the data sharing communities.

## 2. Material and methods

We define reference data as geo-located and time-bound ground truth data that can be either used for training of annual landcover and crop type classification algorithms or for validation of annual landcover and crop type maps. In our search for training and validation data we distinguish 4 different types: 1) Field Observations (FO): on-the-ground observations; 2) Classification or Validation by crowd or expert (CV): check of classified land cover and/or crop type by selected experts or the crowd using auxiliary data such as satellite imagery, street-level pictures, NDVI profiles, usually done in citizen science platforms (e.g. Geo-Wiki and LACO--Wiki); 3) Formal Declaration (FD): formal registration of crop type by individual farmers in support of agricultural policies; and 4) Automated Classification (AC): existing landcover and/or crop type maps. We limited our search for the years 2016 onwards because of WorldCereal's choice to work with recent high resolution feature data (Sentinel 1 and 2).

We have developed a modular framework to find, harmonise, evaluate, and publish reference data to make these data available for training and validation and to share these data with the public, specifically the crop mapping community. The framework consists of 4 steps as indicated in Fig 1. For each of the steps we developed specific protocols building upon previous works and consulting the user community (see S1 Table for these protocols).

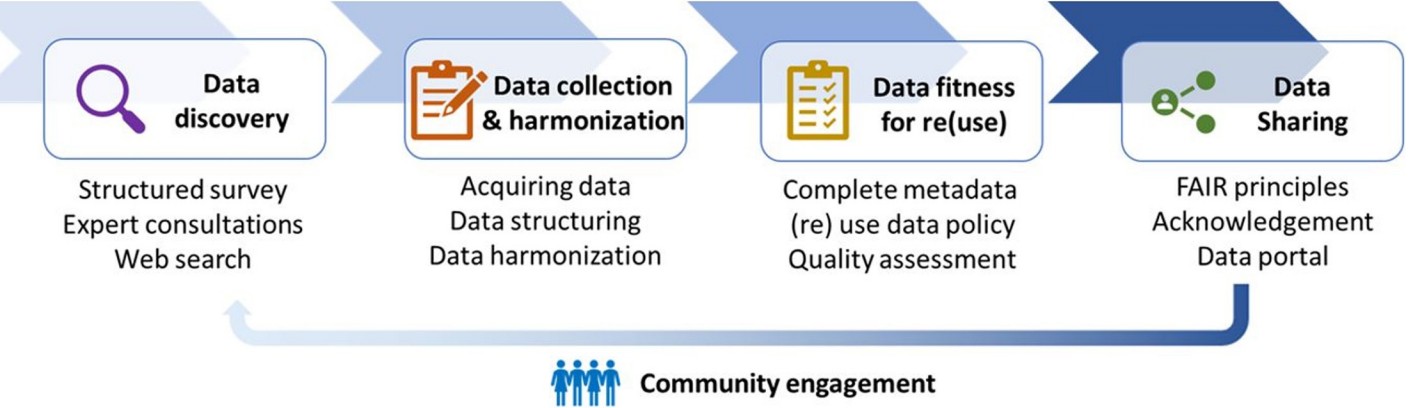

**Fig 1. Modular framework to find, harmonise, evaluate, and publish reference data.**

### 2.1 Data discovery

To discover reference data, we employed several activities such as structured surveys, expert consultations, and web search. A questionnaire was developed in Google Forms, distributed within the crop mapping community, and published on the WorldCereal portal. The questionnaire addressed several aspects of the respondent data sets like the spatial and temporal coverage, data policy, data size, available labels on landcover and crop type etc.

In addition, we organised meetings with the public sector: the European Space Agency (ESA), GEOGLAM, national research institutes hosting JECAM sites, the RadiantEarth foundation, the Future Harvest (CGIAR) centres, NASA Harvest, the Food and Agriculture Organization (FAO), the Copernicus in situ component [23] etc. Similarly, we discussed available reference data with private sector parties active in agricultural input and advisory.

Finally, we performed an electronic database search such as Scopus, Web of Science, Google Scholar. National and international data portals such as GARDIAN were searched. Search queries terms included: "In-situ data", "Reference data", "Crowdsourcing", "Locally based monitoring", "Participatory monitoring", combined with "Crops" or "Cereal Crops Mapping" and more specifically "wheat" and "maize". Literature published from 2016 onwards was considered.

### 2.2 Data collection & harmonisation

The second step involves the downloading, structuring and harmonisation of the data sets using the WorldCereal harmonisation protocol, developed within the project (see S1 Table). From the data inventory we took the data sets fulfilling our criteria i.e., having available geolocation, verifiable timestamp, coming from recent years (2016 onwards), having sufficient ground coverage, and displaying necessary associated metadata such as title, description, creators, licence, and data documentation (citation, licence, contact, location, survey methodology). Data was downloaded either manually or via application programmatic interfaces (API).

Data are prepared in either shapefiles or Geopackages adding the following attributes: landcover (LC), crop type (CT), irrigation (IRR) labels and observation date. In the case of data type CV, additional attributes are required detailing the number of evaluations and the number of people (dis)agreeing. The 'label' attributes are populated by mapping landcover, crop type and irrigation descriptions of the original datasets to the WorldCereal legends. In case of crop type WorldCereal follows the FAO system [24]. For landcover WorldCereal defined the classes as listed in Table 1 [25, 26].

**Table 1. Land cover classes defined in the WorldCereal reference data repository.**

| Name | Value | Description |
|------|-------|-------------|
| No information | 0 | No information |
| Cropland | 10 | Both annual cropland and perennial crop land (under permanent crops). We used this label in case the original label has no indication on annual or perennial cropland. |
| Annual cropland | 11 | Annual cropland is a piece of arable land that is sowed or planted at least once within a 12- months period. Sugarcane plantation and cassava crop are included in this class although they have a longer vegetation cycle and are not yearly planted. |
| Perennial cropland | 12 | Land under permanent crops. |
| Grassland | 13 | Including both temporary and permanent meadows and pastures, with and without grazing. |
| Herbaceous vegetation | 20 | Plants without persistent stems or shoots above ground and lacking definite firm structure. Tree and shrub cover is less than 10%. In WorldCereal this class also includes wetlands. |
| Shrubland | 30 | These are woody perennial plants with persistent and woody stems and without any defined main stem being less than 5 m tall. The shrub foliage can be either evergreen or deciduous |
| Deciduous forest | 40 | Trees with an annual cycle of leaf-on and leaf-off periods. |
| Evergreen forest | 41 | Trees that remain green year round. |
| Mixed/unknown forest | 42 | Mix of deciduous and evergreen forest or unknown. |
| Bare / sparse vegetation | 50 | Lands with exposed soil, sand, or rocks and never has more than 10% vegetated cover during any time of the year. |
| Built up / urban | 60 | Land covered by buildings and other manmade structures. |
| Water | 70 | Lakes, reservoirs, and rivers. Can be either fresh or salt-water bodies. |
| Snow / ice | 80 | Lands under snow or ice cover throughout the year. |
| No cropland (including perennials) | 98 | Other than annual cropland (11). |
| No cropland | 99 | Other than cropland (10, 11, 12). |

Besides the original names we sometimes used additional information e.g., derived from an article associated to the data set, to for example, distinguish between winter and summer cereals if feasible. We also checked the observation date since a proper estimated observation date enhances feature selection, an important step in the training of classification algorithms. This avoids mixing up different seasons e.g., the long and short maize seasons in western Kenya. If a real observation date is missing, we derived the date, although at least the year of observation must be present. For landcover labels, any date within the given calendar year is accepted. In the case of crop type labels, a realistic date representing the middle of the crop season, is estimated combining the given year with local crop calendars. In case of multiple cropping seasons, we also needed the observed season to avoid mixing labels of different seasons. In some cases we used background information to ensure a correct mapping between the year of observation and the calendar year.

## 2.3 Data fitness for (re)use

For a correct and wider (re)use of the publicly available reference data, each data set must be thoroughly described and evaluated. The description of the original data includes the title, information on the owner, the recommended citation, the (re)use data policy, the objective, and aspects of the observation itself like survey method, sampling design and applied quality control. Besides, we described the harmonised data set in terms of space (spatial extent,

resolution, accuracy), time (period covered, accuracy), content (available labels, confidence) and a reference to external documentation explaining the harmonisation.

We developed a set of rules based on existing research [27–29] to assess the spatial, temporal, and thematic accuracy for the different data types to support the selection and use of reference data in the training or validation of crop classification. For each label type (LC, CT, and IRR) a final confidence score was calculated by applying data type specific rules. Three individual scores on spatial, temporal, and thematic accuracy are averaged using specific weights. These weights allow to vary the importance of the three types of accuracy. For example, in the case of data type AC (Automated Classification), we choose to stress the relevance of thematic accuracy as this data type does not hold real observations so thematic accuracy is the biggest concern. All weights and rules are described in detail in the harmonisation protocol (S1 Table).

### 2.4 Data sharing

To make the harmonised data publicly available for wider (re)use, we adhere to the FAIR principles: findability, accessibility, interoperability, and reusability [30]. The reference data are findable and accessible through a common recognized repository and a dedicated portal where users can browse through the different data sets. To support correct re-use each data set has complete metadata including quality assessment, associated data policy of the original data set, suggested citation for proper attribution and a download link to the harmonised data set. In addition, an API and supporting documentation is created within the WorldCereal project.

## 3. Results

### 3.1 Data discovery

The questionnaire was completed by 17 organisations, mainly research institutes responsible for GEOGLAM JECAM sites. In addition, we had responses from CGIAR institutes, NASA Harvest, RadiantEarth foundation and from the private sector OneSoil and the Buenos Aires Grain Exchange. We combined these results with the findings drawn from consulting the crop mapping community. It led to many different leads varying between data from ad-hoc projects to large operational monitoring programs (see S2 Table).

Most data are available via data repositories like CIRAD Dataverse, Joint Research Centre Data Catalogue, Radiant MLHub, Harvard Dataverse, OSF, Figshare, Mendeley data and Github. Other data sets like parcel registration data (LPIS, EUROCROPS) and classified maps (Cropland Data Layer (CDL)–USDA) are offered via dedicated web portals. Some data sets have been published in peer reviewed journals such as the CAWa project, JECAM-CIRAD, INPE-LEM, and LUCAS 2018 Copernicus. Furthermore, some data was shared specifically for the WorldCereal project, for example field survey data provided by the World Food Program (WFP), the Buenos Aires Grain Exchange (BAGE) and the Ministry of Agriculture, Fisheries and Food of Spain (ESYRCE). Note that our inventory was less focused on classified maps as we prefer ground truth data. However, we have added some leads concerning high resolution maps (10–30 m) that could be used for inter-comparison or even training.

### 3.2 Data collection & harmonisation

All leads listed in section 3.1, were scrutinised for selection for harmonisation and use in the WorldCereal reference database. Several data sets were discarded due to various reasons e.g., some data were not sufficiently recent (from 2016 onwards), for example croplands.org, N2Africa, or some others did not include annual crop land, like the Copernicus hotspot LCCE. Other data sets were not available due to restricted data policy e.g., data from some

JECAM sites, CONAB or partially overlap with already existing data e.g., NASA CropHarvest and Eurostat LUCAS, or are still under investigation for ways of sharing. Some data are still in the process of being ingested, e.g., some datasets of Radiant Earth MLHub that were only recently published (Table 2).

Data types AC and FD have 74,805,562 labels (LPIS France, Latvia, Austria, Belgium, EUROCROPS, SIGPAC Spain, Cropland Data Layer (CDL)—USDA) (see Table 3). This high number is mainly due to the full spatial coverage of FD data including most agricultural fields in these countries. The AC data relates to the Cropland Data Layer (CDL)–USDA. It includes 30 m pixels sampled through a fixed spatial scheme avoiding spatial autocorrelation and only selecting pixels with a confidence of at least 80%. The high number of labels of FD and AC is an important asset for training and validation, but it has a strong geographic bias covering western Europe and USA.

Geometry can be either point, polygon, or raster. This distinction is important as polygon and raster datasets allow the training of more advanced algorithms e.g. convolutional neural networks which can not be done using only point datasets.

The reference database includes 987,525 labels of datatype FO and CV (see Table 3). While FO-data originates from many different sources, CV-data comes from Digital Earth Africa only. Around 60% (593,820 labels) are available for the public of which most are points. The public FO and CV data are unequally distributed over the world (see Fig 2 [31]). The observations spread over Africa, Central Asia, and Latin America come from a wide variety of sources and required a substantial effort to harmonize compared to the data from other sources (FD and AC).

Labels of the public FO and CV data mainly cover agricultural areas (cropland, annual cropland, and perennial cropland; 75%) and especially annual crop land (52.7%) (see Fig 3). Around 6% are wheat labels of which 68% relate to winter wheat and only 11% to spring wheat. Wheat observations are concentrated in northern America and Europe. The spring wheat observations are mainly located in the northern part of the USA and western Europe. The share of grain maize labels is about 16% which is substantially larger than wheat. These can be found on the four continents: North and South America, Europe, and Africa.

## 3.3 Data fitness for use

All data under datatype AC and FD have a data policy allowing sharing while acknowledging the owner. Datatype FO has both private and public data (see Table 3). Some data sets were shared for internal use only and thus are stored in the private section of the reference database. It concerns data from ESA project Sen2Agri, BAGE, INTA-field-data, LISTA-field-data, and ESYRCE-Spain Other datasets have policies allowing the use by third parties, possibly limited to only the public sector and they might demand attribution and/or sharing adaptations under the same terms. These datasets are stored in the public section of the reference database while strictly following the license terms. Finally, some data sets have been labelled "private" because we are still investigating possibilities to make the data public. It concerns data sets from NASA Harvest Ukraine and WFP-field-survey. All public data sets have a complete metadata description including the data policy of the original data set.

Confidence scores were calculated for both private and public reference datasets according to the rules defined in the harmonisation protocol (see S1 Table). The summary of the average confidence scores for CT and LC mapping classification is presented in Table 4.

The AC data set has an average confidence score of 66% for CT and 78.5% for LC mapping. The lower score for CT is because of lack of season information, only the calendar year is known. The FD data sets are derived from government institutes. It has the highest average

**Table 2. Data sets that were collected and harmonised (data sharing policy "private" means only shared for internal use within the WorldCereal project).** See S2 Table for more background on these data sets and S3 Table on the recommended citation and data licence.

| Name data set | Countries covered | Years covered | Data sharing policy |
|---|---|---|---|
| AAFC Crop Inventory | Canada | 2016–2021 | Public |
| BAGE (Buenos Aires Grain Exchange) | Argentina | 2018–2020 | Private |
| CAWa project | Uzbekistan, Tajikistan | 2016–2018 | Public |
| CGIAR-CIMMYT | Tanzania | 2019 | Public |
| CGIAR-GARDIAN | Cameroon, Nigeria, India | 2017–2018 | Public |
| COPERNICUS-GEOGLAM | Tanzania, Uganda, Kenya | 2021 | Public |
| Digital Earth Africa | West Africa | 2019 | Public |
| ESA project Sen2Agri | Mali, South Sudan, South Africa | 2016–2017 | Private |
| ESYRCE-Spain | Spain | 2019–2021 | Private |
| EUROCROPS | Germany (partially), Estonia, Lithuania, Portugal, Sweden, Slovenia | 2021 | Public |
| EUROCROPS | The Netherlands | 2020 | Public |
| FAO-WAPOR | Lebanon, Ethiopia, Niger, Egypt, Kenya, Rwanda, Sudan, Mozambique, Senegal, Sri-Lanka | 2017–2021 | Public (except Kenya and partially Lebanon) |
| INPE-LEM | Brazil | 2020 | Public |
| INTA-field-data | Argentina | 2019 | Private |
| JECAM-CIRAD | Brazil, Burkina Faso, Madagascar, Senegal, and South Africa | 2016–2019 | Public |
| JECAM site—Ukraine | Ukraine | 2019 | Private |
| LISTA-field-data | Argentina | 2017 | Private |
| LPIS | Latvia | 2019, 2021 | Public |
| LPIS | France | 2017–2020 | Public |
| LPIS | Belgium | 2017–2019, 2021 | Public |
| LPIS | Austria | 2017–2021 | Public |
| LUCAS 2018 Copernicus | EU countries | 2018 | Public |
| NASA Harvest—Ukraine | Ukraine | 2018–2019 | Private |
| NASA Harvest–CropHarvest[1] | Mali, Kenya, Ethiopia, Zimbabwe, Rwanda, Sudan, Togo, Brazil | 2018–2020 | Public |
| Radiant MLHub | Tanzania, Uganda, Kenya | 2017–2019 | Public |
| OneAcreFund-MEL | Kenya, Rwanda, Tanzania | 2016–2019 | Public |
| OSF-AfSIS | Tanzania | 2017–2019 | Public |
| SIGPAC | Spain | 2018–2019 | Public |
| WFP-field-survey | Uganda, Mozambique, South-Sudan, Malawi, Nigeria, Iraq | 2017–2021 | Private |
| Cropland Data Layer (CDL)—USDA | USA | 2019 | Public |

confidence score of 100% for LC mapping and 94% for CT mapping also because of a lack of season information. The CV data originates from Digital Earth Africa (West Africa) only. Our rules give it a score of 100% because the year is known, the geometry is based on HR-imagery and validation of the reference data samples scores 96.3% (according to the DEA portal [32]).

Datatype FO is most heterogeneous as it has the greatest number of data sets. The data category has an average confidence score of 86.5% for crop type mapping and 95.3% for land cover mapping. Fig 4 depicts a breakdown of the FO scores for CT mapping in the spatial, thematic, and temporal dimensions. The spatial accuracy of these datasets refers to the GPS location inaccuracies, as well as the spatial context (e.g., was the observation properly witnessed within the field or from a nearby road). The spatial accuracy has a mean of 93.3% and the standard deviation of 6.8%. Thematic correctness is 92.0% on average, with a standard deviation of 9.8%. The dataset's average temporal accuracy for crop type was 93.8% with a standard

**Table 3.** Distribution of observations (in %) by data policy (public vs private), geometry (point vs polygon) and year of observation for datatype FO and CV (top) and datatype AC and FD (bottom).

| Datatype FO and CV (987,525 labels) | | | | | |
|---|---|---|---|---|---|
| Year | Public | | Private | | Total |
| | Point | Polygon | Point | Polygon | |
| 2016 | 11.5 | 1.1 | 0.1 | 1.3 | 13.9 |
| 2017 | 9.6 | 0.7 | 0.4 | 0.4 | 11.1 |
| 2018 | 14.6 | 0.8 | 0.4 | 0.9 | 16.7 |
| 2019 | 8.3 | 1.0 | 0.4 | 1.8 | 11.5 |
| 2020 | 4.8 | 0.7 | 0.2 | 17.1 | 22.8 |
| 2021 | 4.2 | 3.0 | 0.0 | 16.8 | 24.0 |
| Total | 53.0 | 7.3 | 1.5 | 38.3 | 100.0 |
| **Datatype AC (raster centres) and FD (polygons) (74,805,562 labels)** | | | | | |
| 2016 | - | - | - | - | - |
| 2017 | - | 16.7 | - | - | 16.7 |
| 2018 | - | 25.5 | - | - | 25.5 |
| 2019 | 1.6 | 26.9 | - | - | 28.5 |
| 2020 | - | 17.7 | - | - | 16.7 |
| 2021 | - | 11.5 | - | - | 4.7 |
| Total | 1.6 | 98.4 | - | - | 100.0 |

deviation of 12.0%. The latter standard deviation is relatively large because we do have a mixed set of observations: some have a real date and others have only a year. In the latter case we give a substantial penalty. The accuracies for LC mapping are similar except for the temporal accuracy which is 100%. All data sets have information on the calendar year which is sufficiently accurate for annual land cover mapping.

### 3.4 Data publication

The harmonised data of the WorldCereal reference database is made available for the public under the license of the original data sets. This includes 593,820 public labels of datatype FO

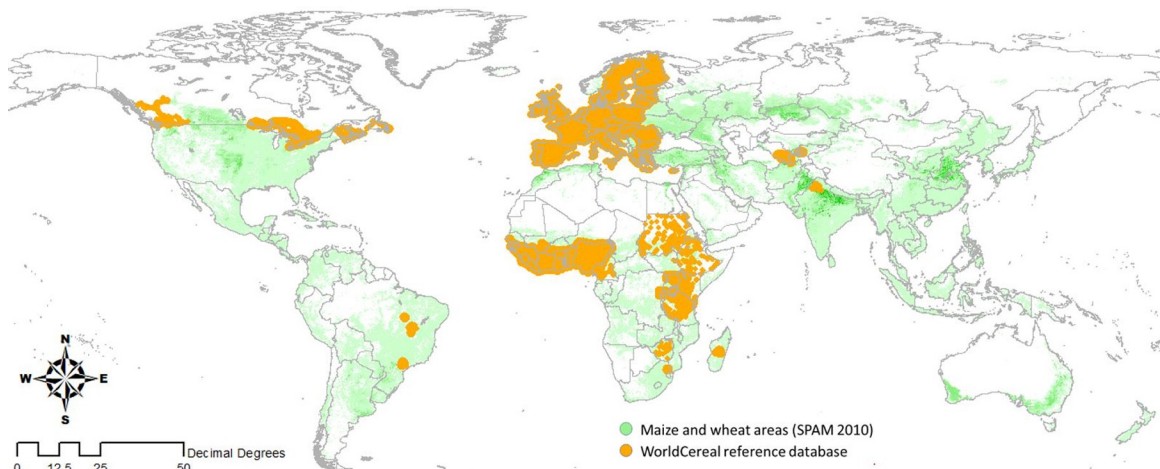

**Fig 2. Spatial coverage of the public FO and CV data, holding 593,820 labels, in the WorldCereal reference database (in orange, lines indicate country borders and green colour indicates main maize and wheat areas according SPAMM 2010 version 2).**

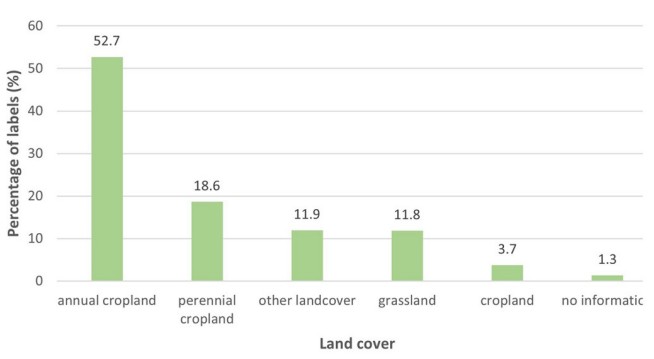
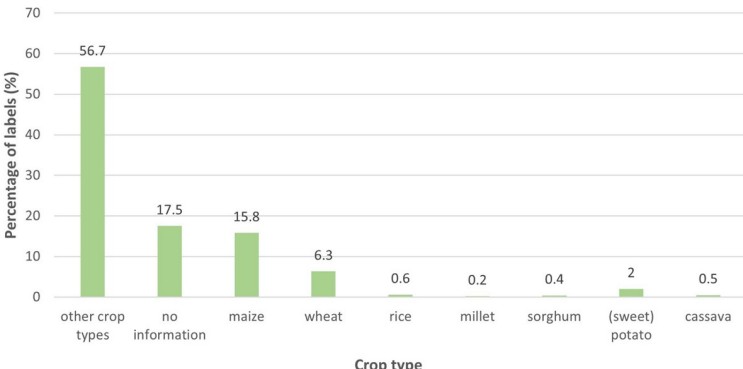

**Fig 3.** Distribution of LC (a) and CT (b) labels of the public FO and CV data (numbers are percentages of total numbers of labels).

and CV and 74,805,562 labels of datatype AC and FD (see Table 3). The harmonised public data are available in Zenodo. All data sets have complete metadata, including citation and data license (see S3 Table) to ensure correct re-use and provide proper acknowledgement to the data owners. To facilitate data exploration and download, we also offer an online open platform with an interface (see Fig 5 and https://worldcereal-rdm.geo-wiki.org).

## 4. Discussion

Within WorldCereal we have built a community-based open harmonised reference data repository at global extent ready for model training or product validation. Data from 2016 onwards were collected from many different sources, harmonised, and annotated. The data sets' spatial, temporal, and thematic quality were assessed. Currently, the repository holds around 75 million harmonised observations with standardised metadata of which most are available to the public. A large share, originating from LPIS data sets, covers only specific countries in Western Europe and is thus biased towards the agro-ecological conditions of these areas. We put substantial effort in the acquisition and harmonisation of data from other regions resulting in around 1 million labels spread over Africa, Latin America, Central Asia, and other countries in Europe of which 60% is public and can be shared. In the following paragraphs we discuss the main findings and issues of collecting, harmonising, evaluating, and publishing reference data.

### 4.1 Data discovery

Despite the work done in WorldCereal still large spatial gaps occur on most continents: Latin America, Africa, Central and Eastern Asia, India, and Australia. Moreover, the data is unequally distributed over the years and crops. Finally, we found very little data on rainfed and irrigation management. Obviously more data exist but is not published for various reasons like lack of urgency, resources and/or because of restrictive data policies. Some researchers

**Table 4. Confidence scores of all data sets (private and public).**

| Type of datasets | Number of datasets | Average confidence score | |
|---|---|---|---|
| | | Crop type | Landcover type |
| Field Observation (FO) | 110 | 86.5 | 95.3 |
| Classification or Validation by crowd or expert (CV) | 4 | | 100 |
| Automated Classification (AC) | 1 | 66 | 78.5 |
| Formal Declaration (FD) | 20 | 94 | 100 |

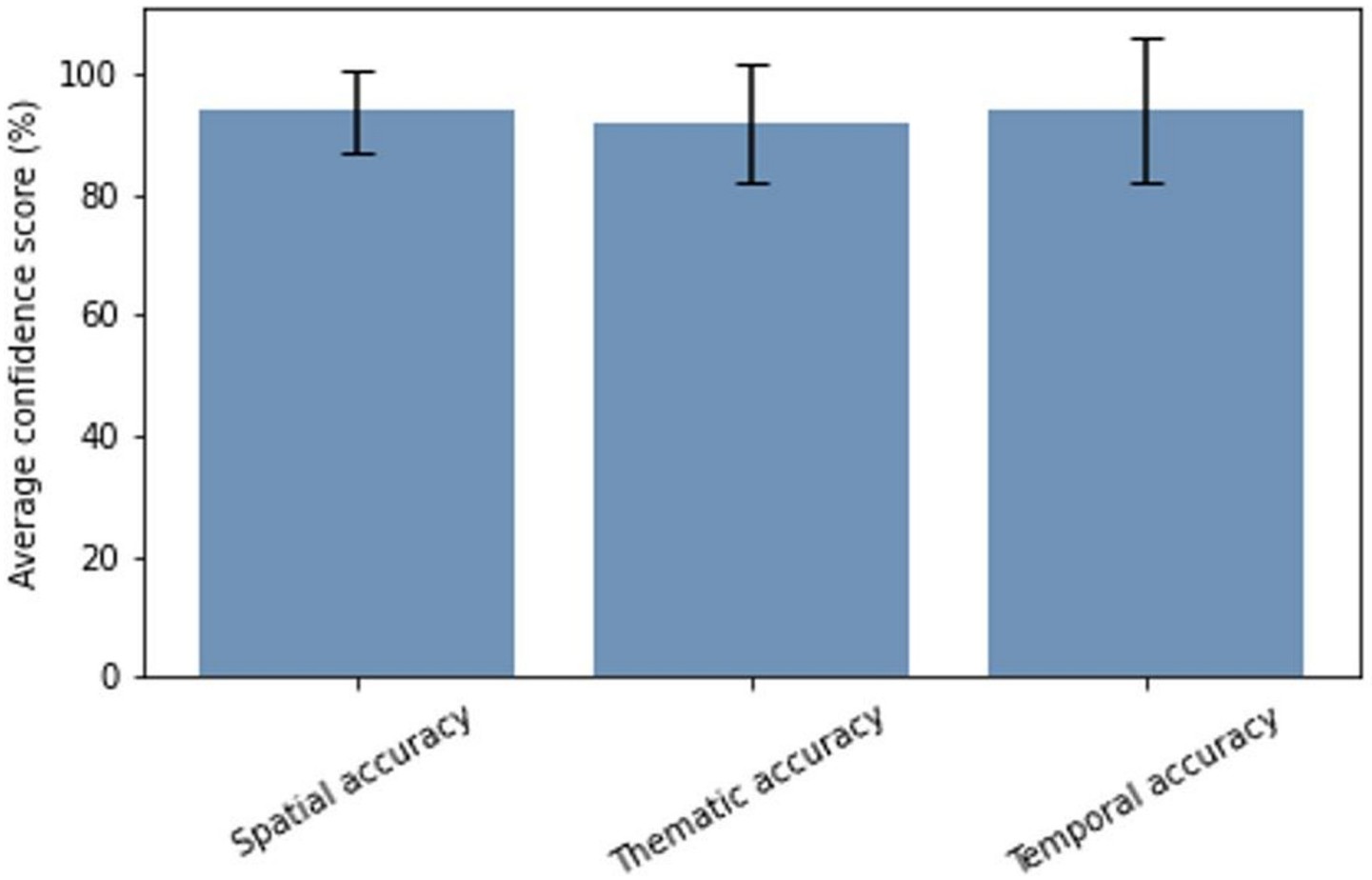

**Fig 4. Breakdown of confidence scores over the different criteria for FO datatype crop type mapping.**

and data owners are reluctant to make their data discoverable and to share their data publicly since they see a competitive advantage by not disclosing any information about their data. However, this will probably change because more funding agencies require the sharing of data when collected with their funds, of course given the legal and privacy context.

To keep the inventory of sources up to date we recommend a community wide joint follow-up to inventory sources we missed and to co-ordinate future acquisition and harmonisation of reference data. To serve a broader suite of mapping products future acquisition campaigns could consider including other aspects of land use such as field size.

While for certain local applications the reference data repository could be rather complete, global mapping initiatives will need to define approaches to deal with gaps in time and space [33]. The same applies to the crop distribution which, for most regions in our repository, does not reflect the actual situation. So downstream users need to take measures e,g. resampling techniques to ensure a sufficient classification.

## 4.2 Data collection and harmonisation

**Harmonising** of data sets requires sufficient metadata and expertise in the agricultural domain. When mapping crops to the WorldCereal legend sometimes multiple options exist as the chosen legend in WorldCereal divides crops on their use. For instance, the label

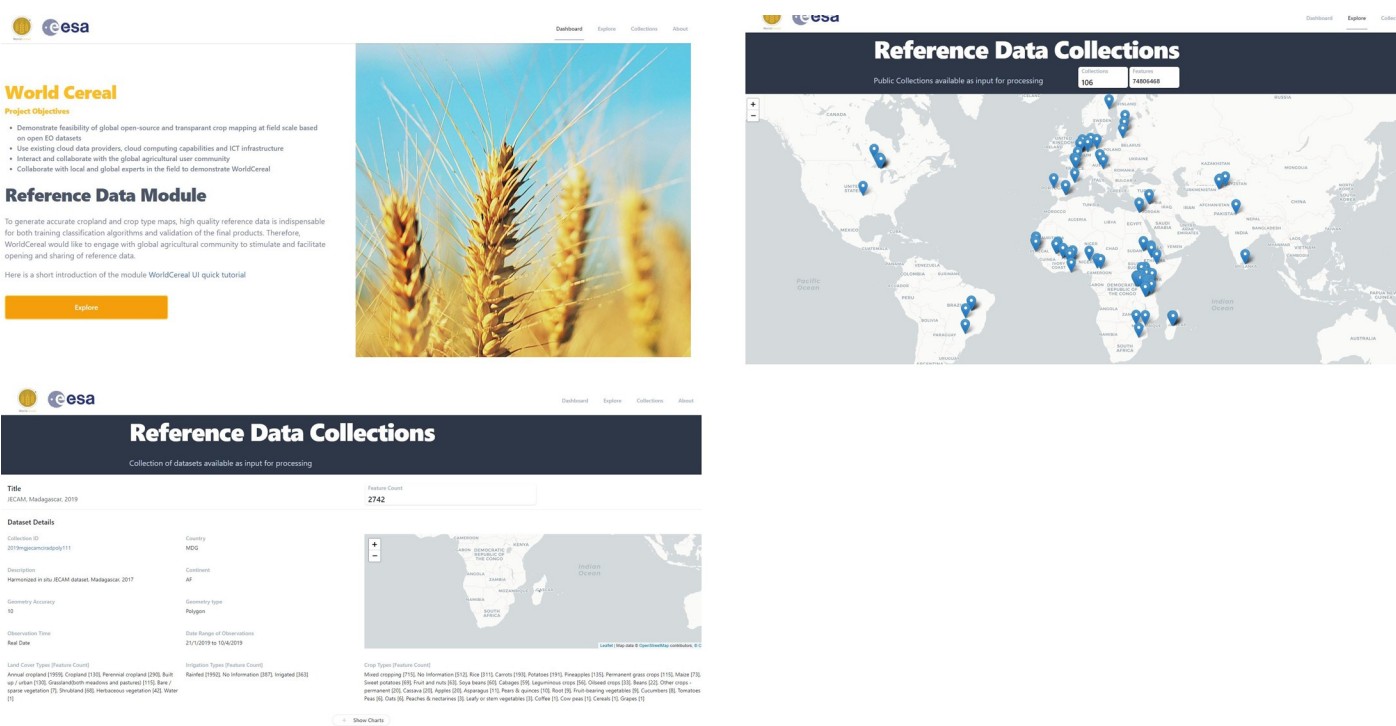

**Fig 5.** Screenshots of the WorldCereal data portal showing the landing page (a), a global view of publicly available data (b) and the details and associated metadata of each contributed data, including the spatial coverage of the data shown on the map (c).

'flax_linseed' can be mapped to CT 'Linseed' (oil seed crops) or CT 'Flax, hemp and similar' (fibre crops). This distinction in crop use, oil or fiber, is less relevant for CT-mapping and therefore labels of these two groups can be combined in training and validation.

If available, we used additional information to map a crop to the most detailed hierarchic level. For example, to indicate winter or spring types in case only wheat was given, we looked for evidence if the survey only covered winter or spring observations. The mapping and underlying assumptions were documented and added to the metadata.

For a correct interpretation and use of the data, the provenance of the individual data sets e.g., survey guidelines are very important. As an example, the OSF-AfSIS data set indicates the presence of a selected set of crops for a location with high precision: 6 decimals lat/lon. The surveyor was instructed to observe crop presence in a 50 m radius circular plot, hence a much lower spatial accuracy compared to the precision of the given co-ordinates. However, in cases only maize was observed (so no presence of any other crop was observed), we could assume only maize was present for that precise location. So, by consulting the survey guidelines we harmonised a data set of maize labels with a high spatial accuracy.

These examples not only illustrate the need for complete metadata including provenance (sources, data cleaning, validation, and quality) but also the effort to do a proper harmonisation making the data ready to use in training or validation of land cover and crop type mapping. This is especially true for legacy data sourced from many different projects and initiatives. Some of the inventoried initiatives have on-going field surveys (e.g., AAFC Crop Inventory, COPERNICUS-GEOGLAM, JECAM, NASA Harvest, WFP, Digital Earth Africa). Here we recommend processing pipelines to map new data to the WorldCereal legends in an efficient and reproducible manner.

### 4.3 Data fitness for (re)use

In this study, we evaluated the fitness for (re)use of datasets in three categories: spatial, temporal, and thematic. Several standards have been proposed in the literature (e.g. OGC standards [34], ISO 19113:2002 [35], Group On Earth Observations (GEO) Data Sharing and Data Management Principles [36]). However, a practical approach for combining these calculations into a single confidence score for LC and CT is lacking. As a result, this is the first attempt to develop the rules and apply it to larger scale reference datasets. These scores help in the selection of a dataset for crop classification training or validation. However, these rules are based on expert advice and have only been tested in WorldCereal. Especially the rule on spatial accuracy needs more attention i.e., applying the rules correctly is labour intensive. For example, we defined a two-step approach to assess the spatial accuracy of arable crops. The first stage is to benchmark observations against non-arable spatial context characteristics such as infrastructure. To do so, we first download the country-specific datasets (e.g., roads, water bodies, railways, buildings, nature areas, and so on) and then do the spatial join operation to ensure that the distance between these features is greater than e.g. 20 m. The second stage is to interpret a sample data set visually using a high-resolution satellite. The approach was tested for a few data sets. A broader assessment of these rules from the crop mapping community is necessary.

In WorldCereal we used the confidence scores to define the weight of an artificial neural network in support of the classification. In the set-up of classification algorithms sample-specific weights determine how much attention the algorithm pays to each sample. This weight is determined to a large degree by the label, but each sample weight eventually gets also multiplied by the confidence score of the harmonised data set. This means that all samples from a data set with more expected issues will get an overall lower weight than one with a higher confidence score.

### 4.4 Data sharing

Many public data were found and harmonised and we encourage the community to continue to share data following the open science and open data principles. To ease the harmonisation process metadata of the published data set should have sufficient information on the provenance especially regarding survey guidelines, post-survey quality control and validation, and basic information on the temporal and spatial aspects.

For data poor areas we also used reference data that was shared with WorldCereal but cannot be shared with the public. In addition, some companies supported WorldCereal by running an in-house validation of the WorldCereal products so that data sharing was not needed. Within the private sector data is rapidly growing [37] but data cannot (easily) be shared. There are initiatives to unlock farmers' data via industry platforms (e.g., api-agro-eu) and data brokers (e.g., Varda) but these are still in an initial phase and could not deliver substantial data yet.

The WorldCereal harmonised reference data repository is available via Zenodo and a GUI interface (available here: https://worldcereal-rdm.geo-wiki.org). On the long term we recommend continuing and institutionalising this reference data initiative e.g., through GEOGLAM, and encouraging the community to publish and share land cover and crop type data. A sustainable continuation of the WorldCereal reference data system will enable the use of APIs to access data and features for a stepwise upload of newly harmonised reference data. Both functionalities are currently not available due to project constraints. The API rest interface will facilitate flexible data retrieval e.g. data for a certain crop or region and can be integrated with other applications or workflows for analysis, storage, and processing. The WorldCereal reference data system also has a step-by-step guided procedure to harmonise and upload data. Currently this upload feature and procedure is not yet operational, but this could be part of the

infrastructure that GEOGLAM is considering building and hosting. To support formatting and harmonisation simple instructions and downloadable documents are available (see S1 Table). This includes mapping to the WorldCereal legends (LC, CT, and IRR codes), metadata templates and a quality assessment of the data. To guarantee a complete and correct publication of data and metadata the procedure defines a manual review by a data steward.

## 5. Conclusion

Within WorldCereal we have built a community-based open harmonised reference data repository at global extent ready for the calibration of image classification deep learning algorithms or the validation of Earth Observation generated products, such as global cropland extent and maize and wheat maps.

Data of 2016 onwards were collected, harmonised, and annotated. Currently, the repository holds around 75 million harmonised observations with standardised metadata of which a large share is available to the public. Still large spatial gaps occur on most continents: Latin America, Africa, Central and Eastern Asia, India, and Australia. Moreover, we found very little data on rainfed and irrigation management.

We developed a set of rules to assess the spatial, temporal, and thematic quality of each data set summarised in one single confidence score. A broader assessment of these rules from the crop mapping community is necessary.

We recommend continuing and institutionalising this reference data initiative e.g. through GEOGLAM, and encouraging the community to publish land cover and crop type data following the open science and open data principles. To keep the inventory of sources up to date we recommend a community wide joint follow-up to inventory sources we missed and to co-ordinate future acquisition and harmonisation of reference data.

.

## Supporting information

**S1 Table. WorldCereal harmonization protocol (see Zenodo 10.5281/zenodo.7584463 for supporting files).**
(DOCX)

**S2 Table. Overview of reference data leads collected in WorldCereal (FO = Field Observation; CV = Classification or Validation by crowd or expert; FD = Formal Declaration; AC = Automated Classification).**
(DOCX)

**S3 Table. Citations and data license of original public data sets harmonized in WorldCereal.**
(DOCX)

## Author Contributions

**Conceptualization:** Hendrik Boogaard, Arun Kumar Pratihast, Steffen Fritz, Sven Gilliams.

**Data curation:** Hendrik Boogaard, Juan Carlos Laso Bayas, Santosh Karanam, Jeroen Degerickx.

**Funding acquisition:** Sven Gilliams.

**Methodology:** Hendrik Boogaard, Arun Kumar Pratihast, Juan Carlos Laso Bayas, Steffen Fritz, Kristof Van Tricht, Jeroen Degerickx, Sven Gilliams.

**Software:** Santosh Karanam.

**Supervision:** Hendrik Boogaard.

**Validation:** Arun Kumar Pratihast, Kristof Van Tricht, Jeroen Degerickx.

**Visualization:** Hendrik Boogaard, Arun Kumar Pratihast, Juan Carlos Laso Bayas, Santosh Karanam.

**Writing – original draft:** Hendrik Boogaard.

**Writing – review & editing:** Hendrik Boogaard, Arun Kumar Pratihast, Juan Carlos Laso Bayas, Santosh Karanam, Steffen Fritz, Kristof Van Tricht, Jeroen Degerickx, Sven Gilliams.

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
