## [Decision Letter · Decision Letter 0]

17 Apr 2023

PONE-D-23-04032Building a community-based open harmonised reference data repository for global crop mappingPLOS ONE

Dear Dr. Boogaard,

Thank you for submitting your manuscript to PLOS ONE. After careful consideration, we feel that it has merit but does not fully meet PLOS ONE’s publication criteria as it currently stands. Therefore, we invite you to submit a revised version of the manuscript that addresses the points raised during the review process.

Discrepancies in datasets for different years and effect on classification; Provide details on the other crops and how they impact the classification accuracy; Provide details on weights of confidence score calculation; Explanations on the imbalances in geographical coverage w.r.t. WorldCereal reference data; Justification on why type of farms or field size not included, etc.

We look forward to receiving your revised manuscript.

Kind regards,

Krishna Prasad Vadrevu, Ph.D

Academic Editor

PLOS ONE

Journal Requirements:

3. Please expand the acronym “ESA” (as indicated in your financial disclosure) so that it states the name of your funders in full.

"The work has received funding from ESA’s WorldCereal project under the project number No. 4000130569/20/I-NB and by the Open Earth Monitor (OEMF) project, a project funded by the European Union's Horizon Europe research and innovation programme under grant agreement No.-101059548."

"The work has received funding from ESA’s WorldCereal project under the project number No.4000130569/20/I-NB and by the Open Earth Monitor (OEMF) project, a project funded by the European Union's Horizon Europe research and innovation programme under grant agreement No.-101059548."

"The work has received funding from ESA’s WorldCereal project under the project number No. 4000130569/20/I-NB and by the Open Earth Monitor (OEMF) project, a project funded by the European Union's Horizon Europe research and innovation programme under grant agreement No.-101059548."

7. We note that you have stated that you will provide repository information for your data at acceptance. Should your manuscript be accepted for publication, we will hold it until you provide the relevant accession numbers or DOIs necessary to access your data. If you wish to make changes to your Data Availability statement, please describe these changes in your cover letter and we will update your Data Availability statement to reflect the information you provide.

8. We note that Figure 2 in your submission contain map images which may be copyrighted. All PLOS content is published under the Creative Commons Attribution License (CC BY 4.0), which means that the manuscript, images, and Supporting Information files will be freely available online, and any third party is permitted to access, download, copy, distribute, and use these materials in any way, even commercially, with proper attribution. For these reasons, we cannot publish previously copyrighted maps or satellite images created using proprietary data, such as Google software (Google Maps, Street View, and Earth). For more information, see our copyright guidelines: http://journals.plos.org/plosone/s/licenses-and-copyright.

(1) You may seek permission from the original copyright holder of Figure 2 to publish the content specifically under the CC BY 4.0 license.  

9. We note that Figures 5-a to 5-c in your submission contain copyrighted images. All PLOS content is published under the Creative Commons Attribution License (CC BY 4.0), which means that the manuscript, images, and Supporting Information files will be freely available online, and any third party is permitted to access, download, copy, distribute, and use these materials in any way, even commercially, with proper attribution. For more information, see our copyright guidelines: http://journals.plos.org/plosone/s/licenses-and-copyright.

(1) You may seek permission from the original copyright holder of Figures 5-a to 5-c to publish the content specifically under the CC BY 4.0 license. 

10.  Please include captions for your Supporting Information files at the end of your manuscript, and update any in-text citations to match accordingly. Please see our Supporting Information guidelines for more information: http://journals.plos.org/plosone/s/supporting-information. 

**Additional Editor Comments:**

Dear Authors,

We have received the comments from two different reviewers. Both suggested minor revisions. Their comments are included for your perusal.

We look forward to the revisions.

Best,

Krishna

Reviewers' comments:

Reviewer's Responses to Questions

**Comments to the Author**

1. Is the manuscript technically sound, and do the data support the conclusions?

Reviewer #1: Yes

Reviewer #2: Yes

2. Has the statistical analysis been performed appropriately and rigorously? 

Reviewer #1: Yes

Reviewer #2: Yes

3. Have the authors made all data underlying the findings in their manuscript fully available?

Reviewer #1: Yes

Reviewer #2: Yes

4. Is the manuscript presented in an intelligible fashion and written in standard English?

Reviewer #1: Yes

Reviewer #2: Yes

5. Review Comments to the Author

Reviewer #1: Review

PLOS One

Title: Building a community-based open harmonised reference data repository for global crop mapping

Manuscript ID: PONE-D-23-04032

Groundtruthing data, also called Reference data, is key to produce reliable crop type and cropland maps. However, collecting the reference data is notoriously challenging, expensive and time-consuming. This paper documents the process to build a community-based, open, harmonised reference data repository at global extent, ready for model training or product validation. The repository holds around 75 million harmonized observations with standardized metadata of which a large share is available to the public (except some which could not be shared due to the restrictions by the original data owners).

This is a huge effort by the authors and their institutes. It is a huge contribution to the community. The data could be used for either the calibration of image classification deep learning algorithms or the validation of Earth Observation generated products, such as global cropland extent and maize and wheat maps. We should than the World Cereal project which is funded by European Space Agency (ESA).The paper is well-written and the repository is accessible. I would recommend acceptance with minor revisions. I listed my (mostly) minor comments below:

1、 In Table 2, the collected data sets covered different years. The land cover types, particularly crop types usually change from year to year and sometimes the changes are considerable. If the datasets are used for classification, how to deal with the time/year difference? How to use these old samples for in-season classification for the current year?

2、 In Figure 3b, the percentage of other crop types is 60.5%. What are other crop types? The percentages of crop types are very imbalance. Will this affect the classification accuracy?

3、 Confidence score is very important for data fitness. Three individual scores on spatial, temporal, and thematic accuracy are averaged using specific weights. These weights allow to vary the importance of the three types of accuracy. How to calculate the weights of confidence score?

Reviewer #2: The WorldCereal project has done a great work of collecting, harmonizing and sharing reference data for crop mapping at the global scale. The quality of the data has been assessed following several criteria and the supply of a confidence score together the data is really appreciated. Overall the paper is well written and presents in an understandable and transparent way the worklow. I have some minor comments below :

1) The WorldCereal reference database is strongly unbalanced in terms of geographical coverage (UE mainly). Even if it is a bit discussed, more discussion is needed on that point and particularly on what implication for global mapping (which is the main purpose of this database)?

2) In terms of information provided in the database, there is the land cover, crop type and irrigation status. I was wondering why information such as the agroecological systems or the type of farms (smallholder, medium, large, cf. Samberg 2016) or the field size (Fritz 2015) have not been included since it could help to refine crop mapping?

3) L143. Table 1. How the crops such as sugarcane or cassava that can have a crop cycle that can last up to 24 months have been considered? From your Table 1, annual cropland is for a period of 12 months but can we considered sugarcane or cassava as perennial crops? Not sure.

4) L150-151. How have you dealt with the case of cropping season spanning two calendar year?

5) L226. The Y-axe label of Figure 3 is a bit confused. A total number is not a percentage.

6) L269. Typing error at WorldCereal.

7) L270. In your table 2, there are several data-sets with year starting in 2016. Along your document you mention at many places that only the data from 2017 are collected. This is a bit confusing.

6. PLOS authors have the option to publish the peer review history of their article (what does this mean?). If published, this will include your full peer review and any attached files.

Reviewer #1: No

Reviewer #2: No

---

## [Author Response · Author response to Decision Letter 0]

2 Jun 2023

We sincerely thank the reviewers for taking the time and effort for reviewing our manuscript “Building a community-based open harmonised reference data repository for global crop mapping”. We heartily appreciated the comments that fundamentally confirmed the importance and relevance of this research. The table is the document named "Response to Reviewers" summarizes our responses to each comment. We have revised our research paper in the light of these useful suggestions and comments and strongly believe that this revision has improved the paper.

---

## [Decision Letter · Decision Letter 1]

14 Jun 2023

Building a community-based open harmonised reference data repository for global crop mapping

PONE-D-23-04032R1

Dear Dr. Boogaard,

We’re pleased to inform you that your manuscript has been judged scientifically suitable for publication and will be formally accepted for publication once it meets all outstanding technical requirements.

Kind regards,

Krishna Prasad Vadrevu, Ph.D

Academic Editor

PLOS ONE

Additional Editor Comments (optional):

Reviewers' comments:

Reviewer's Responses to Questions

**Comments to the Author**

1. If the authors have adequately addressed your comments raised in a previous round of review and you feel that this manuscript is now acceptable for publication, you may indicate that here to bypass the “Comments to the Author” section, enter your conflict of interest statement in the “Confidential to Editor” section, and submit your "Accept" recommendation.

Reviewer #1: All comments have been addressed

2. Is the manuscript technically sound, and do the data support the conclusions?

Reviewer #1: Yes

3. Has the statistical analysis been performed appropriately and rigorously? 

Reviewer #1: N/A

4. Have the authors made all data underlying the findings in their manuscript fully available?

Reviewer #1: Yes

5. Is the manuscript presented in an intelligible fashion and written in standard English?

Reviewer #1: Yes

6. Review Comments to the Author

Reviewer #1: It is a great paper, and I had mostly minor comments with minor revisions. You addressed them well. Congratulations!

7. PLOS authors have the option to publish the peer review history of their article (what does this mean?). If published, this will include your full peer review and any attached files.

Reviewer #1: No

---

## [Editor Report · Acceptance letter]

4 Jul 2023

PONE-D-23-04032R1 

Building a community-based open harmonised reference data repository for global crop mapping 

Dear Dr. Boogaard:

I'm pleased to inform you that your manuscript has been deemed suitable for publication in PLOS ONE. Congratulations! Your manuscript is now with our production department. 

Kind regards, 

on behalf of

Dr Krishna Prasad Vadrevu 

Academic Editor

PLOS ONE